# Temporal dynamics analysis reveals that concurrent working memory load eliminates the Stroop effect through disrupting stimulus-response mapping

**Yafen Li[1], Yixuan Lin[2], Qing Li[3], Yongqiang Chen[3], Zhifang Li[2], Antao Chen[4]***

[1]School of Psychology, Shanghai University of Sport, Shanghai, China; [2]School of Psychology and Cognitive Science, East China Normal University, Shanghai, China; [3]Faculty of Psychology, Southwest University, Chongqing, China; [4]Brain Health Institute, National Center for Mental Disorders, Shanghai Mental Health Center, Shanghai Jiao Tong University School of Medicine and School of Psychology, Shanghai, China

**\*For correspondence:**
antao.chen@sjtu.edu.cn

## eLife Assessment

This **important** study investigates how working memory load influences the Stroop effect from a temporal dynamics perspective. **Convincing** evidence is provided that the working memory load influences the Stroop effect in the late-stage stimulus-response mapping instead of the early sensory stage. This study will be of interest to both neuroscientists and psychologists who work on cognitive control.

**Abstract** Concurrent verbal working memory task can eliminate the color-word Stroop effect. Previous research, based on specific and limited resources, suggested that the disappearance of the conflict effect was due to the memory information preempting the resources for distractors. However, it remains unclear which particular stage of Stroop conflict processing is influenced by working memory loads. In this study, electroencephalography (EEG) recordings with event-related potential (ERP) analyses, time-frequency analyses, multivariate pattern analyses (MVPAs), and representational similarity analyses (RSAs) were applied to provide an in-depth investigation of the aforementioned issue. Subjects were required to complete the single task (the classical manual color-word Stroop task) and the dual task (the Sternberg working memory task combined with the Stroop task), respectively. Behaviorally, the results indicated that the Stroop effect was eliminated in the dual-task condition. The EEG results showed that the concurrent working memory task did not modulate the P1, N450, and alpha bands. However, it modulated the sustained potential (SP), late theta (740–820 ms), and beta (920–1040 ms) power, showing no difference between congruent and incongruent trials in the dual-task condition but significant difference in the single-task condition. Importantly, the RSA results revealed that the neural activation pattern of the late theta was similar to the response interaction pattern. Together, these findings implied that the concurrent working memory task eliminated the Stroop effect through disrupting stimulus-response mapping.

## Introduction

It is critical to effectively deal with the conflict between task-relevant and task-irrelevant information to successfully complete goal-directed behaviors. One of the most commonly employed experimental paradigms for evaluating conflict processing is the color-word Stroop task, which is characterized by lower accuracy and/or longer reaction times (RTs) in incongruent trials compared to congruent trials (*Stroop, 1935*). Of note, an interesting phenomenon is that concurrent verbal working memory task can reduce or even eliminate the Stroop conflict (*Kim et al., 2005*; *Luna et al., 2020*; *Park et al., 2007*; *Zhao et al., 2014*). In terms of the load theory based on multiple resources, cognitive resources are limited while specific to different cognitive processes. When substantial resources are allocated to maintaining memory information, the resources required to process same-dimension distractors are occupied, reducing the interference caused by the distractors (*Brockhoff et al., 2022*; *Luna et al., 2020*; *Park et al., 2007*). Although this theory attempts to explain the disappearance of the Stroop conflict, its description remains relatively vague, and it does not further reveal which specific processes are influenced during Stroop conflict processing.

Stroop conflict processing should be understood not as a single process but as multiple successive subprocesses, including stimulus processing, stimulus-response mapping, and response output stage (*Banich, 2019*; *Parris et al., 2022*; *Viviani et al., 2024*). Related research suggests that there might be two possibilities. The first one is that the concurrent working memory task primarily decreases the neural encoding of perceptual visual features, known as the early-stage modulation hypothesis. The sensory recruitment hypothesis posits that the sensory cortex is crucial for maintaining visual content by modulating sensory gain, thereby biasing processing toward relevant stimuli (*Hallenbeck et al., 2021*; *Wolff et al., 2020*). Moreover, several researchers have suggested that the Stroop conflict relies more on the input-driven allocation of attention to perceptual features (*Algom et al., 2022*; *Bugg and Crump, 2012*; *Haciahmet et al., 2023*; *Schmidt, 2019*). Accordingly, these views suggest that working memory maintenance competes for neural resources with the lower-level visual encoding of distractors, hindering conflict formation at an early stage. Besides, the second one is that the concurrent working memory task primarily weakens higher-level processing, termed the late-stage modulation hypothesis. The theoretical underpinning is that the working memory system comprises not only sensory buffers but also a central system (*Baddeley, 2012*) that mainly recruits the prefrontal cortex, partially overlapping with the brain networks involved in Stroop conflict processing (*Friehs et al., 2020*; *Okayasu et al., 2023*). Furthermore, some research indicated that filtering out highly familiar and easily processed semantic information during the early sensory processing stage is difficult for individuals (*Augustinova et al., 2019*; *Cavanagh and Frank, 2014*; *Itthipuripat et al., 2019*). Consistent with this hypothesis, both relevant and irrelevant sensory inputs receive comprehensive semantic processing, while working memory load constrains the process of binding higher-level semantic codes with responses, potentially influencing the subsequent response output stage. Overall, the two hypotheses, although distinct, are not mutually exclusive and can be inspected from a time-dynamic perspective.

In the present study, a single task (the classical manual Stroop task) and a dual task (the Sternberg working memory task combined with the Stroop task) were employed to explore the aforementioned hypotheses with electroencephalography recording of high temporal resolution. Several event-related potentials (ERPs) highly relevant to Stroop conflict processing will be focused on in this study. Previous studies have proposed that within cognitive control mechanisms, occipito-parietal P1 signifies the earliest stage of top-down processes in early sensory processing, gating the direction of information processing in the brain (*Klimesch et al., 2007*). Additionally, numerous studies have identified two ERP components exclusively elicited by the Stroop task (i.e. the N450 and the sustained potential [SP]). N450 is a central-parietal negative deflection that exhibits greater negativity in incongruent trials compared to congruent trials within 350–500 ms after stimulus onsets, which is typically interpreted as conflict monitoring, the process of evaluating conflict levels and communicating the conflict information to control systems (*Coderre et al., 2011*; *Heidlmayr et al., 2020*; *Xu et al., 2024*). As for the SP component, it shows a centro-parietal positive deflection from approximately 600 to 1000 ms in incongruent conditions, which is regarded as an index of conflict resolution (*Di Russo and Bianco, 2023*; *Donohue et al., 2016*; *Heidlmayr et al., 2020*; *Vo et al., 2021*). In all, these ERP components, which are distinctly time-sequential, can help explore the dynamic effects of working memory on Stroop conflict processing.

In addition to the ERP components, conflict processing also involves various neural oscillatory activities. First, extensive research has found that fronto-central theta (4–7 Hz) power increases in conflict situations with the time windows of early and late theta band effects in stimulus-locked analyses corresponding to those of the N450 and SP effects, which suggests that early and late theta power reflects conflict detection and resolution processes, respectively (*Yaar-Soffer et al., 2024*; *Senoussi et al., 2022*; *Xu et al., 2024*; *Zhao et al., 2015*). Second, the occipital-parietal alpha oscillations (8–13 Hz) can serve as a neural inverse index of mental activity or alertness, associated with the attentional engagement with visual stimuli during conflict processing. The decreased alpha activity in the incongruent condition indicates enhanced attentional suppression (*Arakaki et al., 2022*; *Tafuro et al., 2019*; *Zhou et al., 2023*; *Zhu et al., 2023*). Third, previous study has shown that the response incongruent trial evokes a small central beta power than the stimulus incongruent trial in the Stroop task (*Zhao et al., 2015*). The suppressed effect of the central beta power most likely reflects the inhibition of proponent responses, associated with motion preparation (*Böttcher et al., 2023*; *Chung et al., 2024*; *Wendiggensen et al., 2022*; *Zhao et al., 2015*). Accordingly, these frequency bands can provide a comprehensive characterization of the Stroop conflict processing.

It is worth realizing that neural oscillations are not characterized by a predetermined time window like ERPs. However, multivariate pattern analysis (MVPA) can help to solve the difficulty of hypothesizing which frequency band or the time window is changed in the Stroop task. As a data-driven technique, MVPA provides novel insight into the differences between two conditions by identifying the features that contribute most to classification (*Li et al., 2023*; *Li et al., 2024*; *Lin et al., 2023*; *Meier et al., 2012*). Therefore, the present study will combine time-frequency analysis and MVPA to inspect the dynamic neural characteristics of the modulation of working memory on Stroop conflict processing at the neural oscillatory level.

Notably, even if differences are found in ERPs or neural oscillations, the extent to which they are linked to behavior is an interesting question. Representational similarity analysis (RSA) is applied to build a connection between behavior and neural activities in the present study (*Kriegeskorte and Kievit, 2013*). Recent studies have suggested that RSA is a reliable way to uncover the neural mechanisms behind conflict processing (*Freund et al., 2021*; *Yang et al., 2023*), which helps identify which neural activities are central to the influence of working memory on Stroop conflict processing. Overall, the present study employs electroencephalography (EEG) technology, through MVPA and RSA, in an attempt to comprehensively uncover the dynamic neural mechanisms underlying the elimination of the Stroop effect influenced by working memory when the dimensions of memory information and distractors overlap.

## Results

### Behavioral results

For behavioral analysis, a two-way repeated measures analysis of variance (ANOVA) (task type: single task vs. dual task×congruency: congruent vs. incongruent) was performed on RTs. The results showed that the RTs of the single-task condition were faster than the dual-task condition (main effect of task type: $F_{(1, 31)}$=113.28, p<0.001, $\eta^2_p$=0.79) and a reliable Stroop interference effect (main effect of congruency: $F_{(1, 31)}$=32.53, p<0.001, $\eta^2_p$=0.51). Notably, the interaction between task type and congruency was significant ($F_{(1, 31)}$=8.62, p=0.006, $\eta^2_p$=0.22). Simple effect analysis revealed that, in the single-task condition, the RTs of incongruent trials were significantly slower (M=851.15, SE = 32.16) than congruent trials (M=736.10, SE = 26.29) (p<0.001). However, in the dual-task condition, the RTs did not differ between incongruent trials (M=1073.04, SE = 39.73) and congruent trials (M=1033.68, SE = 49.43) (p=0.089), indicating that the Stroop effect of RTs only occurred in the single-task condition (*Figure 1*).

### ERP results

#### P1

Two-way repeated-measures ANOVA was conducted on the P1 component, revealing a significant main effect of task type ($F_{(1, 31)}$=29.92, p<0.001, $\eta^2_p$=0.49). Specifically, the P1 amplitude of the dual task (M=1.63, SE = 0.30) was significantly greater than the single task (M=0.39, SE = 0.28). However,

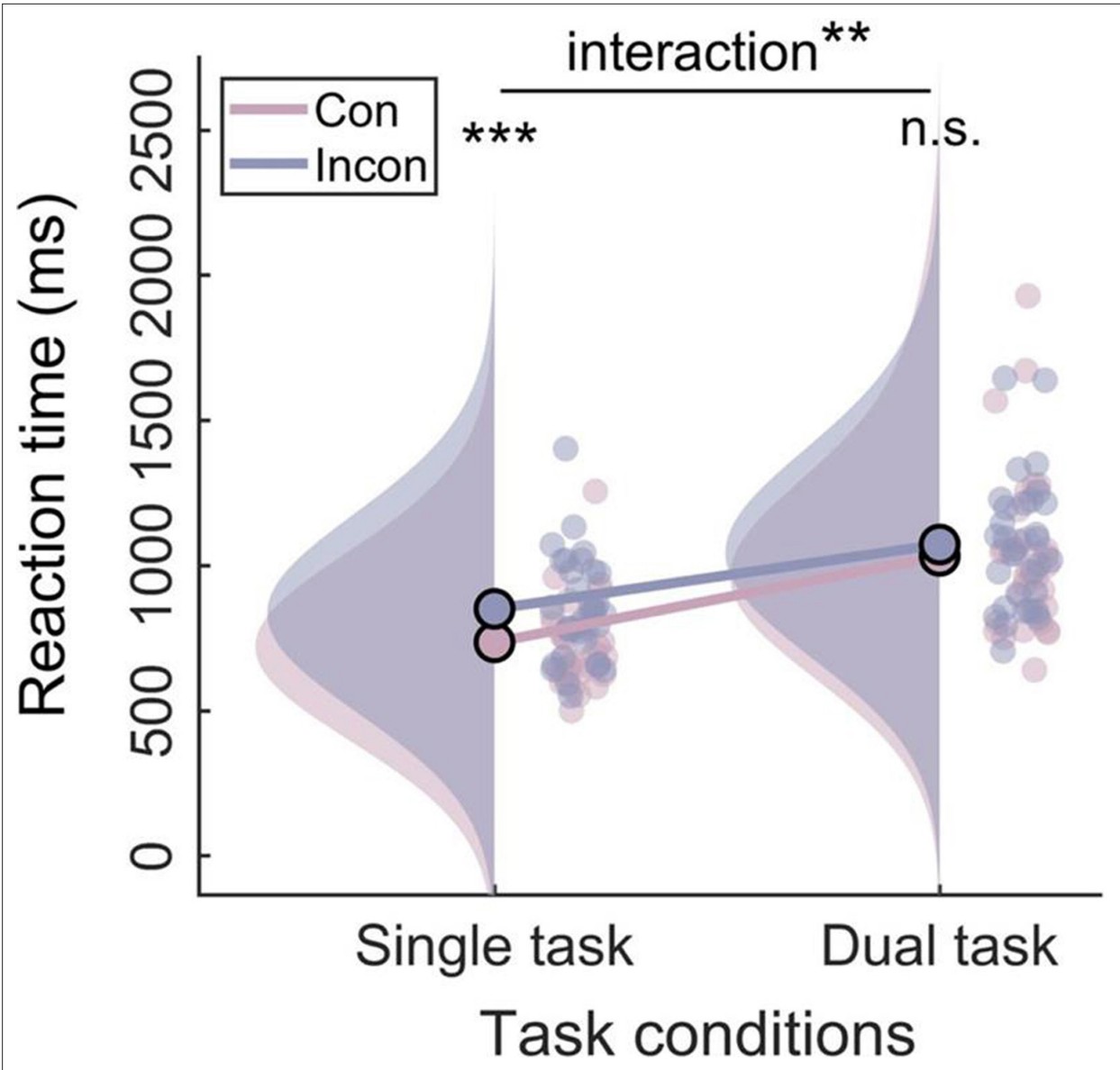

**Figure 1.** Raincloud plots (**Allen et al., 2019**) of behavioral data. The plot consists of a probability density plot, a line graph, and raw data points. **p<0.01, ***p<0.001.

neither the main effect of congruency ($F_{(1, 31)}$=0.00, p=0.99, $\eta^2_p$=0.00) nor the two-way interaction ($F_{(1, 31)}$=0.23, p=0.63, $\eta^2_p$=0.007) was significant (**Figure 2**).

### N450

Analysis of the N450 component showed the main effect of task type ($F_{(1, 31)}$=14.71, p=0.001, $\eta^2_p$=0.32) and the main effect of congruency ($F_{(1, 31)}$=4.36, p<0.05, $\eta^2_p$=0.12) were both significant. However, the two-way interaction ($F_{(1, 31)}$=0.63, p=0.43, $\eta^2_p$=0.02) was not significant (**Figure 3**).

### SP

For the SP component, two-way repeated-measures ANOVA showed the main effect of task type was significant ($F_{(1, 31)}$=14.54, p=0.001, $\eta^2_p$=0.32), but the main effect of congruency was not ($F_{(1, 31)}$=2.89, p=0.10, $\eta^2_p$=0.09). Importantly, the interaction between congruency and task type was significant ($F_{(1, 31)}$=5.68, p<0.05, $\eta^2_p$=0.16). Simple effect analysis showed that, in the single task, the SP amplitude in incongruent trials (M=1.29, SE = 0.39) was significantly greater than in congruent trials (M=0.70, SE =

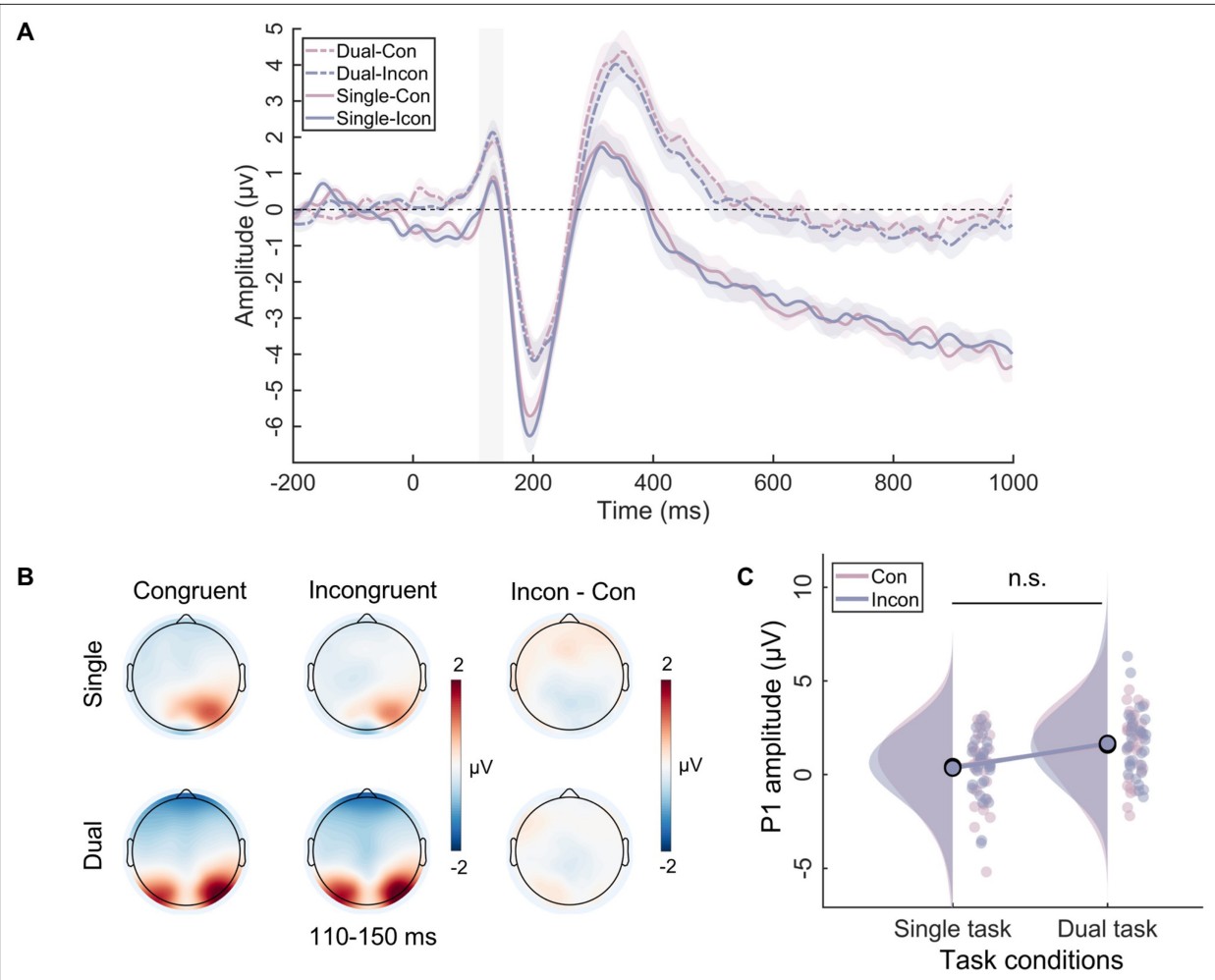

**Figure 2.** The P1 results in the single task and the dual task. (**A**) The event-related potential (ERP) waveforms for the P1 component. The shaded rectangle represents the defined time windows, and the shading above and below the ERP waveform represents the 95% confidence intervals. (**B**) Topographic distributions of average amplitude for each condition within the time window (110–150 ms) and the amplitude differences between congruent and incongruent trials in single and dual tasks. (**C**) Raincloud plots of amplitude for P1 across conditions. n.s., not significant.

0.30) (p=0.009). On the contrary, in the dual task, the SP amplitude did not differ between congruent trials (M=0.27, SE = 0.34) and incongruent trials (M=0.22, SE = 0.37) (p=0.98) (*Figure 4*).

## MVPA results

Cluster-level permutation tests of the theta band indicated that only one cluster in 740–820 ms could significantly classify the Stroop effect of the single and dual tasks (*Figure 5A*). Regarding the beta band, tests showed a significant above-chance difference between the two task types from 920 to 1040 ms (*Figure 5C*). However, no reliable decoding was observed between the two task types in the alpha band (*Figure 5B*).

## Time-frequency results

### Theta band

The two-way repeated-measures ANOVA on theta power revealed that the main effect of congruency was significant ($F_{(1, 31)}$=14.68, p=0.001, $\eta^2_p$=0.32), but the main effect of task type was not ($F_{(1, 31)}$=0.30, p=0.59, $\eta^2_p$=0.01). Notably, the interaction was significant ($F_{(1, 31)}$=7.71, p=0.009, $\eta^2_p$=0.20). Simple effect analysis showed that, in the dual task, the theta power did not differ between congruent trials (M=3.12, SE = 0.24) and incongruent trials (M=3.27, SE = 0.24) (p=0.63). However, in the single task,

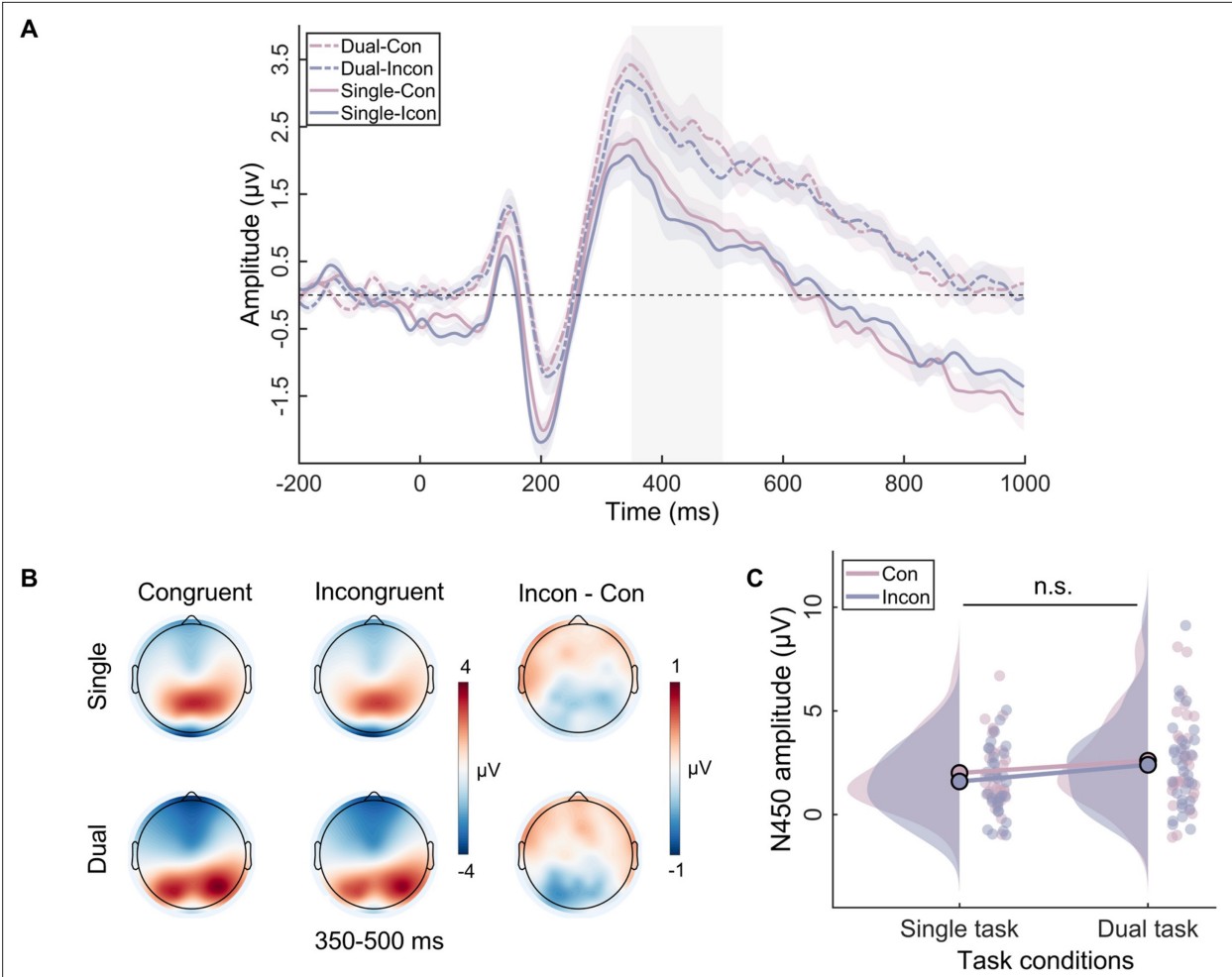

**Figure 3.** The N450 results in the single task and the dual task. (**A**) The event-related potential (ERP) waveforms for N450 components. The shaded rectangle represents the defined time windows, and the shading above and below the ERP waveform represents the 95% confidence intervals. (**B**) Topographic distributions of average amplitude for each condition within the time window (350–500 ms) and the amplitude differences between congruent and incongruent trials in single and dual tasks. (**C**) Raincloud plots of amplitude for N450 across conditions. n.s., not significant.

the theta power was significantly higher in incongruent trials (*M*=4.00, SE = 0.32) than in congruent trials (*M*=2.68, SE = 0.20) (p<0.001) (**Figure 6**).

### Beta band
Analysis of the beta power showed the main effect of task type ($F_{(1, 31)}$=34.55, p<0.001, $\eta^2_p$=0.53) and the main effect of congruency ($F_{(1, 31)}$=11.61, p=0.002, $\eta^2_p$=0.27) were both significant. Importantly, the interaction between task type and congruency reached a significant level ($F_{(1, 31)}$=15.64, p<0.001, $\eta^2_p$=0.34). Simple effect analysis revealed that, in the single-task condition, the beta power was significantly lower in the incongruent trials (*M*=0.53, SE = 0.06) than in the congruent trials (*M*=0.80, SE = 0.08) (p<0.001), while no significant difference was found between them in the dual-task condition (p=0.40) (**Figure 7**).

## RSA results
The RSA results showed that the interaction pattern in the theta band significantly contributes to variation in the interaction pattern of response time (estimate = 0.17, SE = 0.05, *t*=3.22, p~corrected~ = 0.004). However, the correlation between the dissimilarity matrix of the SP (estimate = 0.02, SE = 0.05, *t*=0.48, p~corrected~ = 0.63) and beta (estimate = 0.10, SE = 0.05, *t*=2.03, p~corrected~ = 0.06) representation and the interaction pattern of response time was not significant (**Figure 8**).

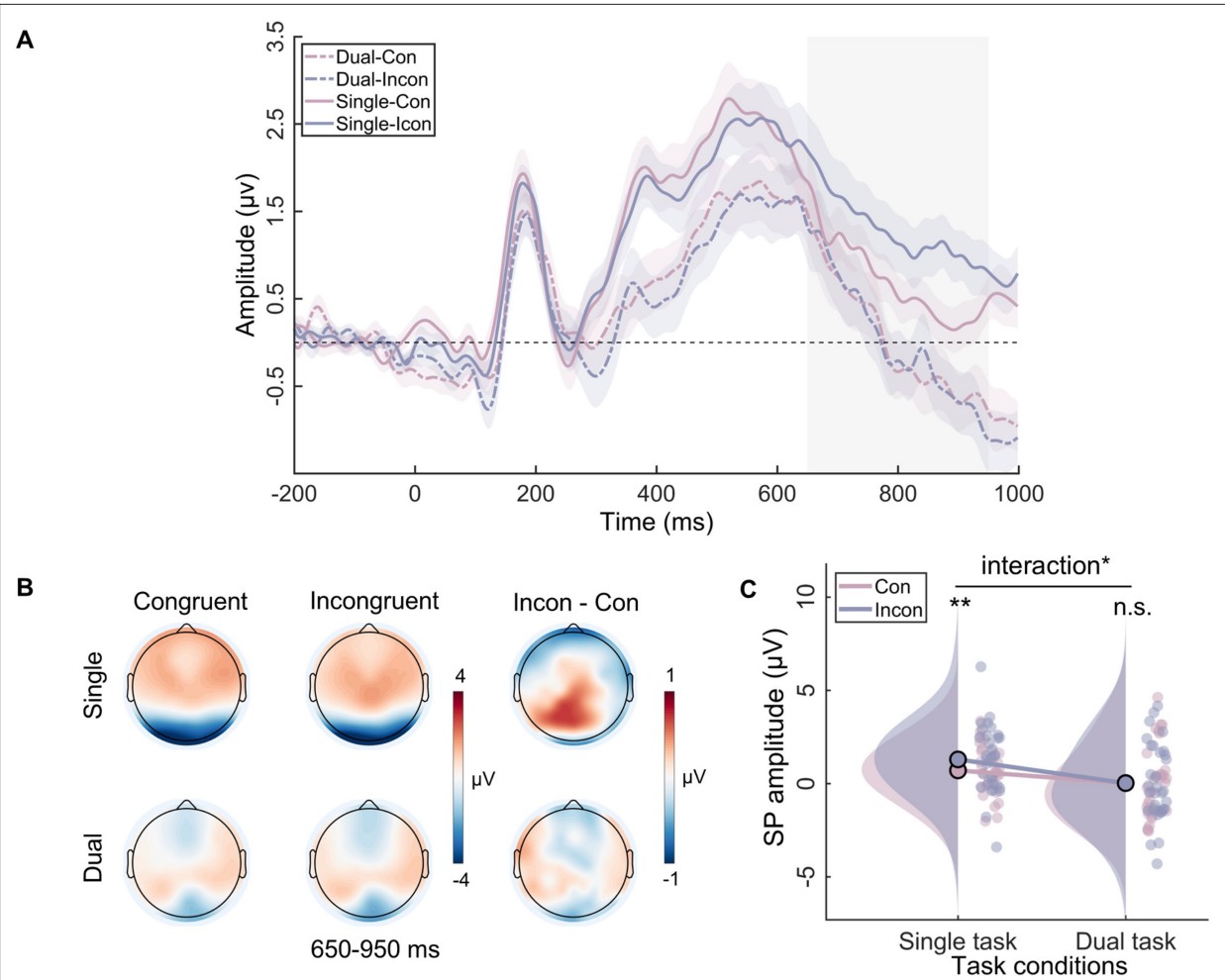

**Figure 4.** The sustained potential (SP) results in the single task and the dual task. (**A**) The event-related potential (ERP) waveforms for SP components. The shaded rectangle represents the defined time windows, and the shading above and below the ERP waveform represents the 95% confidence intervals. (**B**) Topographic distributions of average amplitude for each condition within the time window (650–950 ms) and the amplitude differences between congruent and incongruent trials in single and dual tasks. (**C**) Raincloud plots of amplitude for SP across conditions. n.s., not significant, *p<0.05, **p<0.01.

## Supplementary results

Analysis of the alpha power showed the main effect of task type ($F_{(1, 31)}$=57.90, p<0.001, $\eta^2_p$=0.65) and the main effect of congruency ($F_{(1, 31)}$=7.38, p=0.01, $\eta^2_p$=0.19) were both significant. However, the two-way interaction ($F_{(1, 31)}$=0.53, p=0.47, $\eta^2_p$=0.02) was not significant (*Figure 5—figure supplement 1*).

## Discussion

The present study systematically investigated the dynamic neural mechanisms underlying the modulation of concurrent working memory task on Stroop conflict processing when the dimensions of memory information and distractors overlap. At first, we verified behaviorally that working memory processing eliminated the conflict effect. Specifically, in the single task, the response times of incongruent trials were significantly slower than congruent trials, but there was no significant difference between the two in the dual task. For the ERP results, no significant interaction was found in P1 or N450, but in SP. Namely, it showed that the incongruent trials evoked more positive SP than the congruent trials in the single task, while there was no difference between the two in the dual task. Moreover, MVPA results showed that only the theta power (740–820 ms) and the beta power (920–1040 ms) of conflict effect could successfully classify different task types. Further, in the corresponding time windows, we

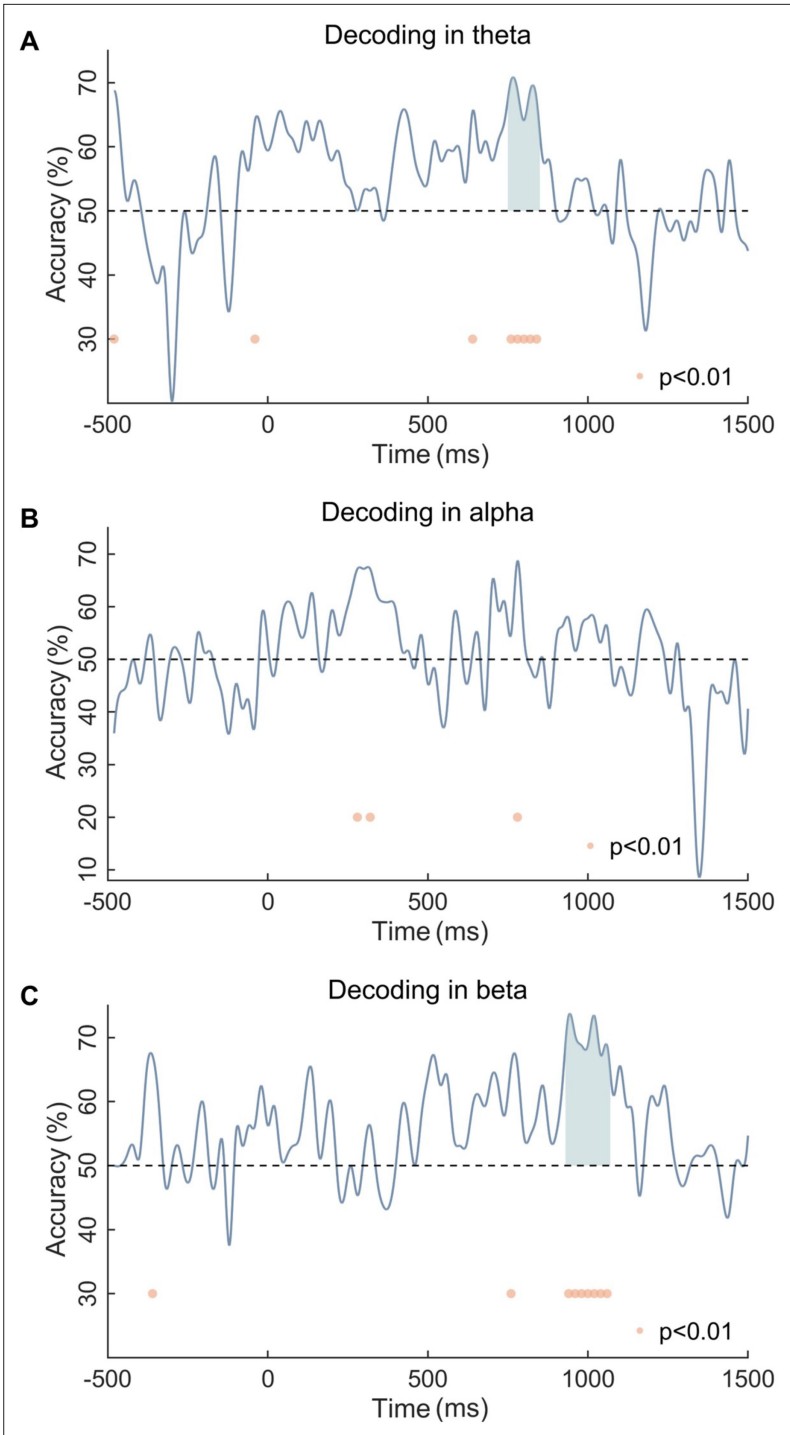

**Figure 5.** Multivariate pattern analysis (MVPA) results for (**A**) the theta band, (**B**) the alpha band, and (**C**) the beta band. Dots denote significant differences from chance (p<0.01). Shading represents clusters of significant time points (cluster-corrected).

The online version of this article includes the following figure supplement(s) for figure 5:

**Figure supplement 1.** Time-frequency results for the alpha band.

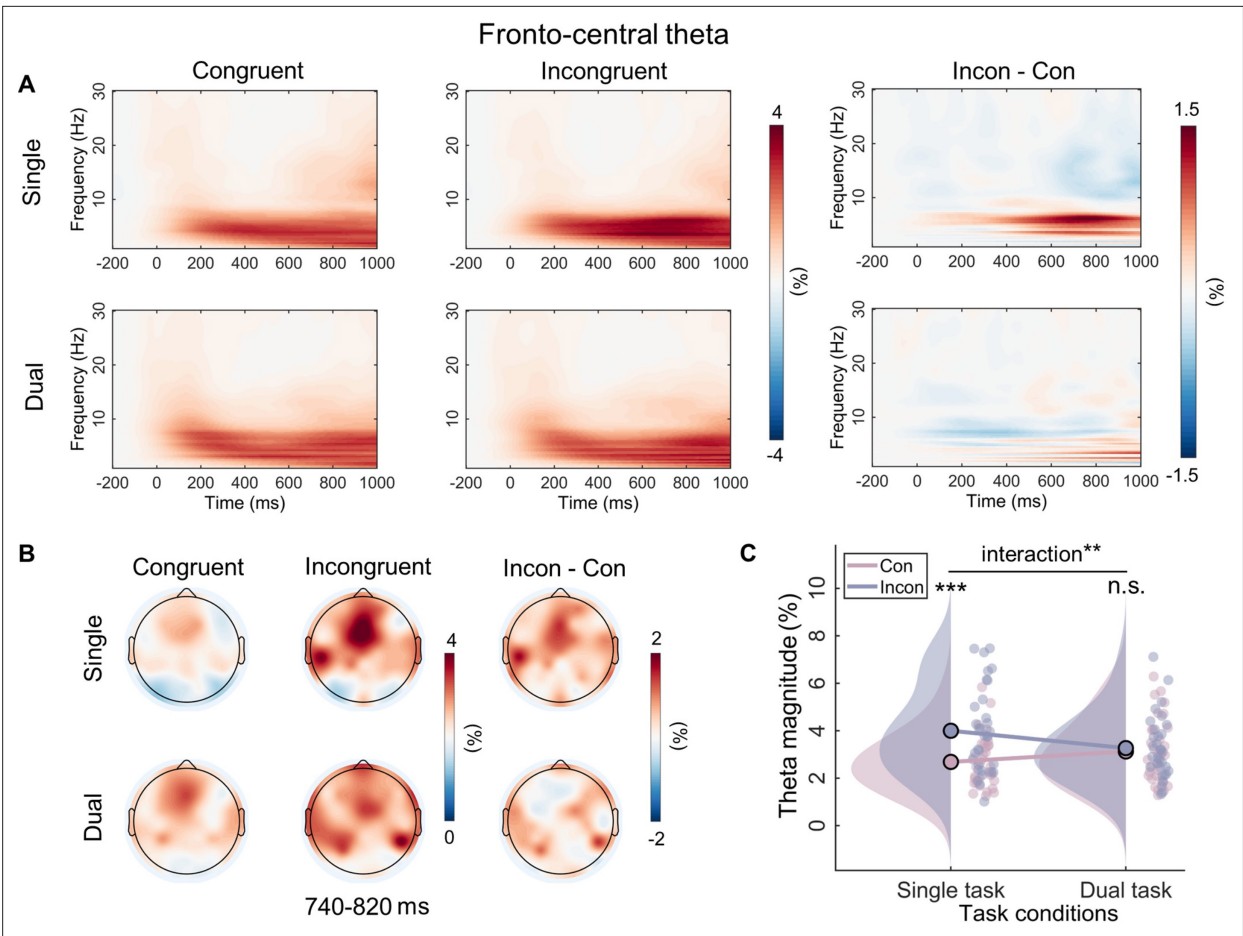

**Figure 6.** Time-frequency results for the theta band. (**A**) Grand-average time-frequency representations for the theta band over fronto-central area (Fz, FCz, Cz) for each condition, and the difference activity between congruent and incongruent trials in single and dual tasks. (**B**) Topographic distributions of average amplitude for each condition within the defined time-frequency region of interest (4–7 Hz, 740–820 ms) and the magnitude differences between congruent and incongruent trials in single and dual tasks. (**C**) Raincloud plots of event-related spectral perturbation (ERSP) magnitudes for theta band across conditions. n.s., not significant. **p<0.01, ***p<0.001.

found that only in the single task, the incongruent trial induced greater theta power and smaller beta power than the congruent trial. Finally, RSA showed that only the interaction pattern in theta power correlated with response time. All these results collectively suggested that, when the memorized information overlaps with the dimension of distractors, working memory processing modulates the late stage rather than the early stage of Stroop conflict processing.

Behaviorally, we found that the Stroop interference effect was reduced under the dual-task condition, replicating previous findings (*Kim et al., 2005*; *Luna et al., 2020*; *Park et al., 2007*; *Zhao et al., 2014*). Consistent with the latest load theory (*Murphy et al., 2016*; *Park et al., 2007*), there are multiple resources, each with its own processing-specific, limited-capacity pool. Working memory load will occupy the limited resources necessary for distractor processing, diminishing the intensity of distractor processing and thereby reducing their interference with the task. In our study, concurrent visual short-term working memory maintenance occupied the semantic resources required for distractor processing, resulting in the elimination of the Stroop effect. Interestingly, a previous study (*Zhao et al., 2014*) that reported a diminished Stroop interference effect employed the classic Chinese character Stroop task as well, but the working memory task involved memorizing meaningless visual shapes made from hand-drawn lines. This seems inconsistent with the load theory hypothesis, as the images appear to occupy spatial-related resources rather than overlapping with the distractor stimulus processing dimension. However, in contrast to English words, the processing of Chinese characters relies more on graphemic encoding and memory (*Chen, 1993*). Therefore, the processing of line patterns occupies some of the resources needed for character processing, which aligns with our

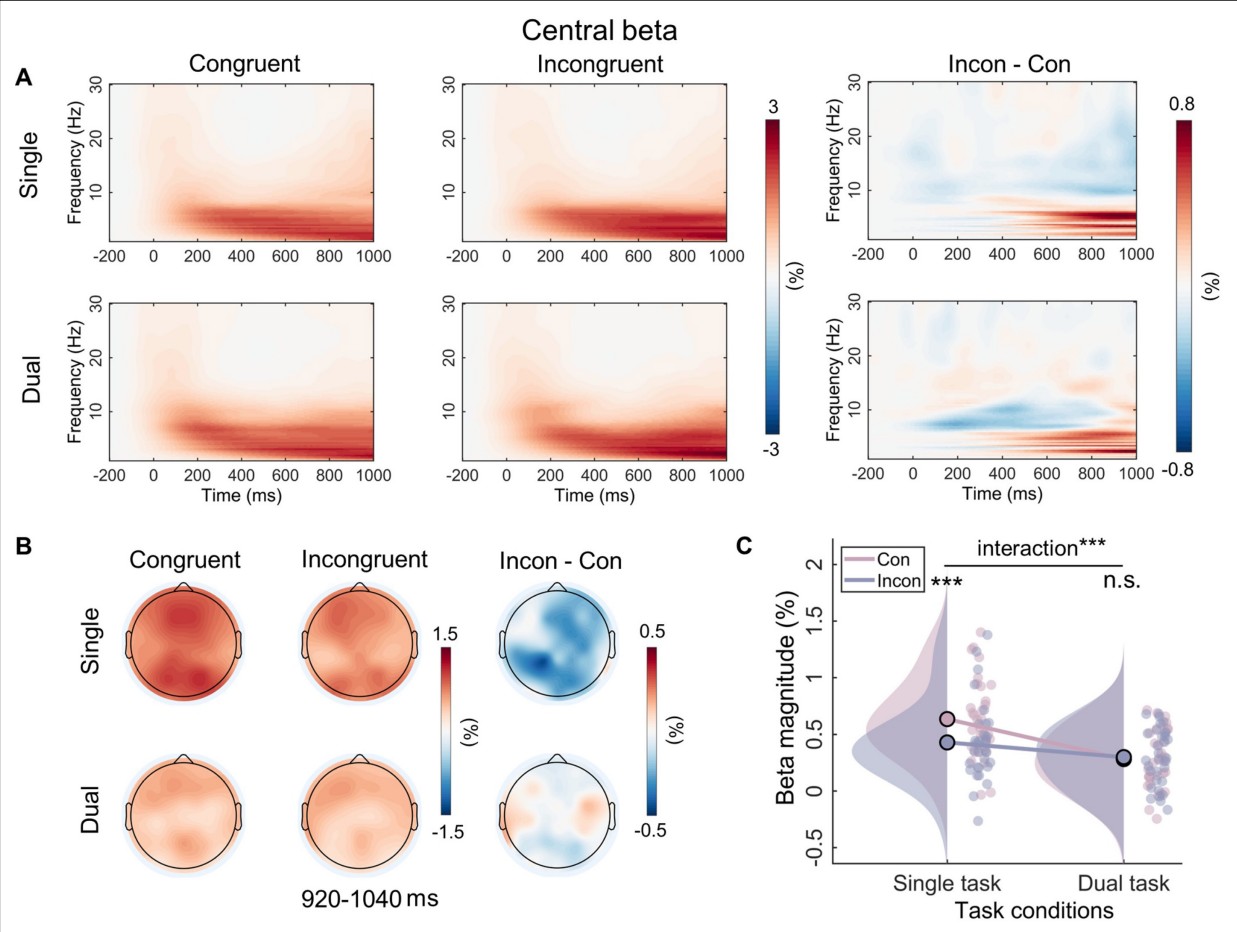

**Figure 7.** Time-frequency results for the beta band.

study's hypothesis based on dimensional overlap. Additionally, this study suggested that the modulation effect of the working memory task on the typical Chinese character Stroop task was relatively minor. We speculate that this is because graphic processing occupies lower-level resources related to the overall shape and spatial structure of Chinese characters (*Zhang et al., 2020*). In contrast, our study, which employed Chinese characters as the working memory load, could directly influence higher-level resources involved in semantic processing. Therefore, we believe that verbal working memory load can significantly modulate the Stroop effect. Overall, the elimination of the Stroop effect observed in our study is consistent with the core hypotheses of load theory based on multiple resources, paving the way for further exploration into the neural mechanisms behind this elimination.

The ERP findings indicated that the working memory task influenced only the SP component, which reflects the process of conflict resolution. Consistent with most research, our study showed that the SP amplitude is more positive in conflict situations (*Heidlmayr et al., 2020*; *Zhao et al., 2015*). Crucially, our study found that verbal working memory load eliminated the difference in SP between congruent and incongruent trials, indicating that conflict resolution did not occur in the dual-task condition. Additionally, our results revealed a congruency effect at N450, indicating that participants indeed detected the occurrence of conflict (*Heidlmayr et al., 2020*). Furthermore, our study did not find any differences in P1 between the different congruency conditions, which aligns with the findings of most studies that indicate the early sensory input processing stage is insensitive to congruency effects (*David et al., 2011*; *Di Russo and Bianco, 2023*). In other words, participants cannot suppress the processing of distractors at the perceptual level. Generally, the overall ERP results suggest that working memory influences the later stage of conflict resolution, while not influencing the early stage of perceptual processing and conflict detection.

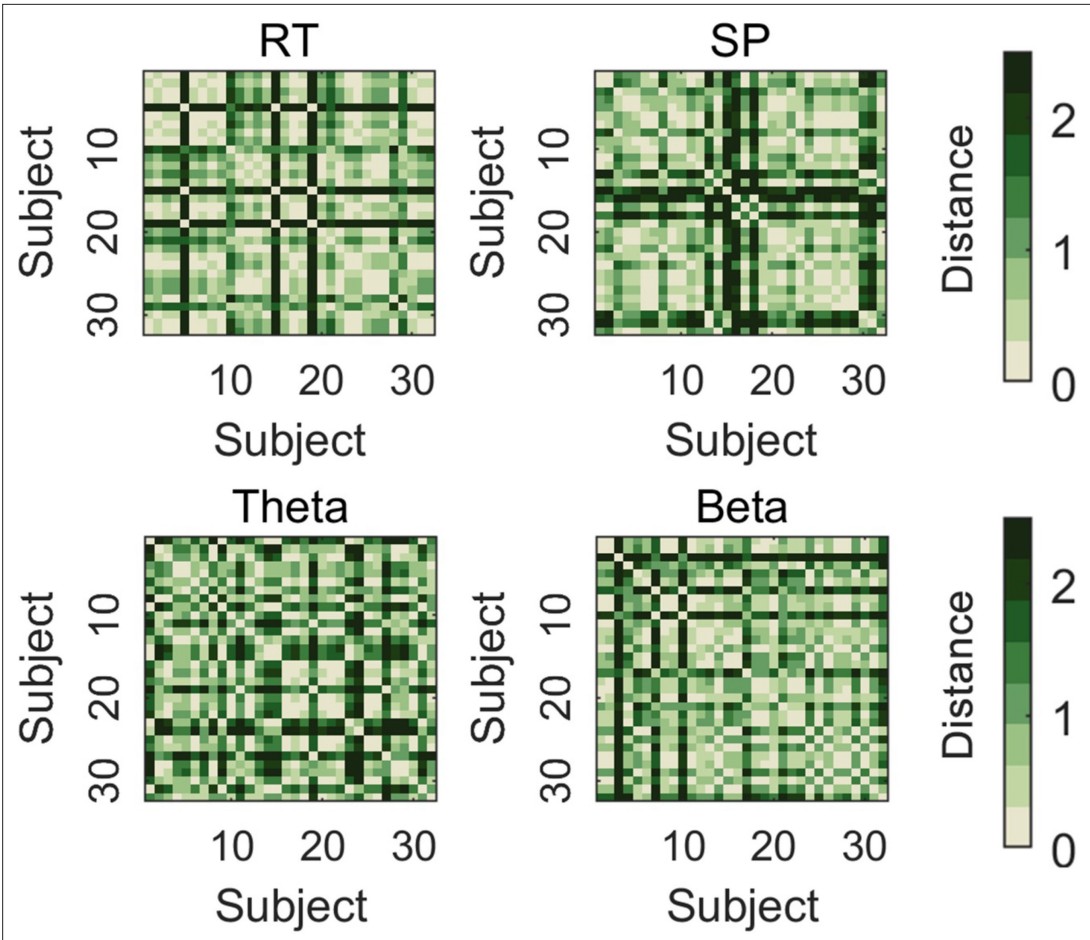

**Figure 8.** Representational dissimilarity matrices (RDMs) for the interaction pattern in response time (RT), sustained potential (SP), theta, and beta. Each RDM is calculated as the Euclidean distances of data reflecting interaction pattern from 32 subjects.

In addition to the ERP results, the MVPA results from time-frequency also provide supportive evidence that working memory modulates Stroop conflict processing in the later stage. First, in line with the previous study (*Zhao et al., 2014*), there are no continuous time periods in which alpha activities (incongruency minus congruency) can successfully discriminate the single task and the dual task. Alpha is regarded as an indicator of top-down attentional modulation, with an increase in alpha reflecting the inhibition of distractor input (*Van Diepen et al., 2019*; *Zhou et al., 2023*), implying that the patterns of attentional adjustment are similar across different task types in our study (*Figure 5— figure supplement 1*). This finding may be related to the attentional rejection template (*Wen et al., 2022*). In the Sternberg task, participants are required to remember only seven Chinese characters and reject the others, which is similar to the attention template used for rejecting characters in the Stroop task. Second, the different task types can be successfully discriminated by late theta activity after the stimulus presentation (740–820 ms). Similarly, previous research has found that current working memory leads to smaller differences in theta activity (400–1000 ms) between incongruent and neutral trials (*Zhao et al., 2014*). According to the framework of event coding theory, late theta may represent the process of stimulus-response binding or mapping (*Beste et al., 2023*; *Wendig- gensen et al., 2023*). And this has been demonstrated in several empirical experiments. For example, theta is thought to be involved in adjusting the processing of stimulus information to correspond with appropriate responses (*Cavanagh et al., 2009*). In line with this opinion, about –200 to 0 ms before a response, higher theta activity is observed in high-conflict conditions compared to low-conflict condi- tions (*Cohen and Donner, 2013*). Accordingly, working memory may eliminate the Stroop effect by influencing the stimulus-response mapping process. Lastly, the present study also shows that beta activity between 920 and 1040 ms after stimulus presentation can successfully discriminate between

the two task types. Beta activity is often thought to be associated with motor priming. Specifically, it decreases at the onset of movement (*Jenkinson and Brown, 2011*) but increases during holding a static status (*Baker et al., 2001*). Since response button selection definitely occurs after conflict resolution and is a late motor adjustment according to the stimulus, this result suggests that working memory may also influence the response output stage of Stroop conflict processing.

The RSA results further reveal that the neural activity pattern of theta is most similar to the behavioral Stroop effect modulation pattern. This result can be interpreted from the standpoint of neural efficiency, suggesting that a reduction in activation patterns reflects greater neural efficiency (*Rypma et al., 2006*). Therefore, we speculate that the core reason for the disappearance of the behavioral Stroop effect under the dual-task condition is that, compared to congruent trials, neural activation in theta decreases and response speed increases in incongruent trials (*Figures 1 and 6C*). Moreover, this result aligns with many studies that have found theta can predict RTs (*Kaiser and Schütz-Bosbach, 2021*; *Senoussi et al., 2022*). However, although SP and beta exhibited interaction effects in the data patterns, they may not be the core indicators directly determining the behavioral modulation effect. SP, as an ERP component, contains multiple oscillatory bands, indicating several parallel brain processes (*Yaar-Soffer et al., 2024*). Moreover, theta oscillations are biological candidates for neuronal computation and communication (*Cavanagh and Frank, 2014*), capable of integrating selection-related information during cognitive control to inform response execution (*Duprez et al., 2020*). Combined with the MVPA decoding results indicating that the beta band (920–1040 ms) follows the theta band (740–820 ms), we speculate that changes in beta bands may be influenced by theta bands. Overall, when the dimensions of working memory load overlap with those of distractor stimuli, individuals do not need to engage in conflict resolution. Specifically, the pre-memorized Chinese characters occupied the resources for the late-stage stimulus-response mapping represented semantically by the distractors, weakening the mapping of distractors to incorrect responses and subsequently inhibiting the response output of the distractors, resulting in the disappearance of response conflict.

Theoretically, this study expands the existing hypotheses of load theory from a temporal dynamics perspective. Furthermore, the results present a new viewpoint for understanding conflict processing, indicating that individuals cannot suppress the processing of irrelevant stimuli at the perceptual level in the early stage but may regulate this processing later on. Importantly, this study indicates that the regulatory platform for this modulation is working memory, highlighting its crucial role in conflict processing. Beyond its role in representing the target, working memory could also be crucial in the suppression of distractors, suggesting a strong interrelationship between the two core components of cognitive control (*Diamond, 2013*). Practically, the effective application of working memory tasks in learning or work environments, by increasing the working memory load overlapping with distracting stimuli, can assist individuals in managing cognitive conflicts more efficiently. Additionally, the findings may serve as a theoretical basis for developing inhibitory control intervention strategies aimed at enhancing working memory to improve inhibitory control.

In summary, this study demonstrated that the concurrent working memory task can eliminate Stroop conflict through disrupting stimulus-response mapping. The results indicated that concurrent working memory load influenced the late stage related to stimulus-response mapping and response selection processes, indexed by SP, late theta band, and beta band, while it did not influence the early stage associated with perceptual processing, conflict detection, and attention adjustment, indexed by P1, N450, and alpha band. RSA further identified theta as the neural indicator directly associated with the behavioral modulation pattern. These findings suggested that when the working memory load overlaps with the dimension of distractor stimuli, it occupies the resources for the stimulus-response mapping of the distractor, weakening the stronger response tendency of the distractor and ultimately leading to the disappearance of response conflict. Accordingly, the present study comprehensively characterized the dynamic neural mechanisms by which working memory modulates Stroop conflict processing, further deepening our understanding of the mechanisms underlying conflict generation.

## Materials and methods

### Participants

Using G*Power software (*Faul et al., 2007*), we calculated the sample size for the experiment. The power analysis (power≥0.9) on within-factors designs, assuming a moderate effect size of 0.25,

indicated a sample size of 30. We recruited 34 healthy subjects (17 females; age: 21.38±1.92 years) through advertisements. Two subjects were excluded from further analyses due to excessive EEG artifacts. Finally, a total of 32 subjects (16 females; age: 21.44±1.97 years) remained for behavioral and EEG analyses. All participants were right-handed, had normal or corrected-to-normal vision, and normal color perception assessed by the Ishihara Color Test, and reported no history of neurologic or psychiatric disease. They all signed written informed consent before the experiment. All procedures were conducted in accordance with the principles of the Declaration of Helsinki and its later amendments and were approved by the Human Research Human Ethics Committee of the Shanghai University of Sport (102772024RT005).

## Apparatus and stimuli

Participants were instructed to perform experiments in a soundproof room. The experimental stimuli were presented on a 19-inch color monitor (resolution 1920×1080, refresh rate 60 Hz) with a viewing distance of approximately 65 cm. Stimulus presentation and response recording were controlled by a Microsoft PC running MATLAB (Mathworks, Inc) with Psychophysics Toolbox extensions (*Brainard, 1997*; *Pelli, 1997*).

The experiment employed the single task (the classical manual Stroop task) and the dual task (the Sternberg working memory task combined with the Stroop task). The Stroop task stimuli consisted of four Chinese words: '红' (red, RGB: 255, 0, 0), '绿' (green, RGB: 0, 255, 0), '蓝' (blue, RGB: 0, 0, 255), and '黄' (yellow, RGB: 255, 255, 0). They were presented in the semantically corresponding font color (congruent trials) or a different font color (incongruent trials) in the center of a black background (visual angle, ~1.15 × 1.15°). The Sternberg working memory task involved seven Chinese characters chosen randomly among 14 familiar Chinese characters: '人' (human), '刀' (knife), '大' (big), '水' (water), '天' (heaven), '口' (mouth), '月' (moon), '立' (stand), '山' (mountain), '上' (up), '手' (hand), '石' (stone), '女' (woman), and '火' (fire) (*Zhao et al., 2010*). They were equally spaced on a horizontal row at the center of the display (visual angle for each character, ~1.15 × 1.15°).

## Procedure

For the dual task (*Figure 9A*), the subjects were required to memorize seven Chinese characters in anticipation of a recognition test at the end of the trial. Each trial began with a white fixation point for a random duration from 400 to 600 ms, followed by seven white Chinese characters that required memorization for 2000 ms. After a jittered 800–1200 ms fixation point, participants performed a Stroop task in which the color of the words was identified as quickly as possible by pressing the corresponding button ('F' with the left index finger, 'D' with the left middle finger, 'J' with the right index finger, and 'K' with the right middle finger), while attempting to ignore the task-irrelevant semantic meaning. The display for the words remained on the screen until a response was made. After the Stroop task, a fixation point was randomly presented for 400–600 ms and terminated by the appearance of a probe character, to which the subject had to determine if the character was part of their memorized list of items. During 50% of the trials, the probe letter was included as part of the memorized set. Participants were asked to answer by pressing one of two response buttons ('S' with the left ring finger and 'L' with the right ring finger). All response buttons in this experiment were

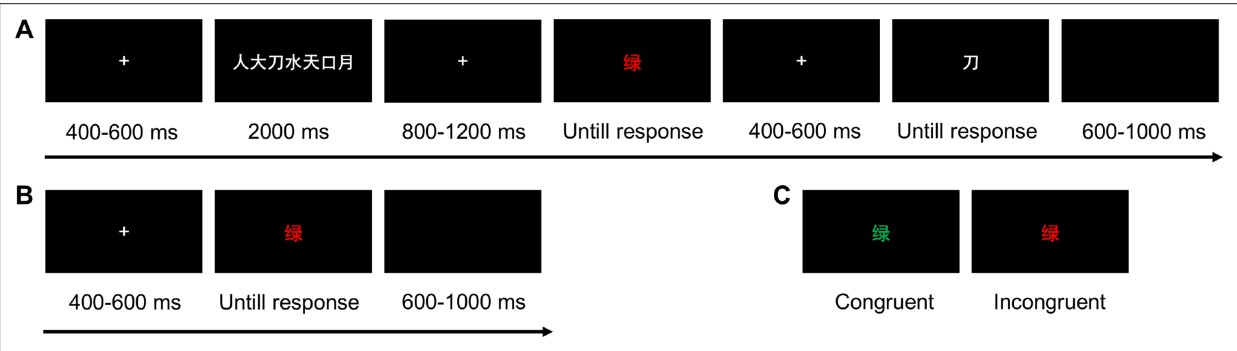

**Figure 9.** Schematic illustration of experimental trial structures and stimuli types. (**A**) Dual-task structure. (**B**) Single-task structure. (**C**) Representative stimuli for congruent and incongruent conditions.

counterbalanced across subjects. After a response was given, the intertrial interval was randomized between 600 and 1000 ms until the onset of the next trial. For the single task (*Figure 9B*), subjects were only required to complete the individual Stroop task. Each block alternated between single and dual tasks in an ABAB fashion, and the task type in the first block was counterbalanced across participants. The subjects first performed practice blocks of 144 trials. Participants were permitted to participate in the formal experiment when their average error rate was below 15%. The formal experiment included six blocks, totaling 144 trials, with an equal distribution of 50% congruent and 50% incongruent trials presented randomly in each block. The entire experiment lasted for around 30 min.

## EEG recording and preprocessing

EEG data were recorded using standard 64 in-cap Ag/AgCl electrodes, following the extended international 10–20 system (Brain Products GmbH, Germany; pass band, 0.01–100 Hz; sampling rate, 1000 Hz). Vertical and horizontal electrooculograms were recorded from below the left eye and outer canthus of the right eye, respectively. The FCz served as an online reference. The impedance of all electrodes was maintained below 5 kΩ during the entire recording process.

The offline EEG data were preprocessed by EEGLAB V2023.0 (*Delorme and Makeig, 2004*) and custom scripts in MATLAB 2021a. First, the continuous data was down-sampled from the original sampling rate of 1000 to 500 Hz. Then, the data were filtered with a 30 Hz low-pass filter and a 0.1 Hz high-pass filter. After that, we re-referenced the data to the average activity of all scalp channels and manually inspected the data to remove obvious artifacts. Subsequently, we carried out an independent component analysis (Infomax algorithm) and thereby identified and manually removed artifacts such as eyeblinks, horizontal eye movements, heart, muscle, and line noise. As a next step, the data was segmented based on the trial type (single-task congruent vs. single-task incongruent vs. dual-task congruent vs. dual-task incongruent) and was time-locked to the Stroop stimulus onset. Continuous EEG data were epoched from −200 to 1000 ms relative to word onset, with a baseline correction applied using the 200 ms pre-stimulus interval. Finally, trials with amplitude values exceeding ±100 μV at any electrode were rejected. Overall, a mean of 12.59% (SD, 4.82%) of all trials were excluded.

## Data analysis

### Behavioral analysis

Only trials that were correct in both the memory task and the Stroop task were included in all subsequent analyses. In addition, trials in which response times deviated by more than three standard deviations from the condition mean were excluded from behavioral analyses. Overall, a mean of 4.8% (SD, 3.31%) of all trials were excluded. We performed repeated-measures ANOVA with Greenhouse-Geisser correction if needed, in which the within-subject factors were congruency (congruent, incongruent) and task type (single task, dual task). Adjustments for multiple comparisons were realized using Bonferroni correction.

### ERP analysis

Considering previous studies (*Coderre et al., 2011*; *Xu et al., 2024*; *Zivony and Lamy, 2022*), P1 was calculated using a time window of 40 ms (110–150 ms) at bilateral occipito-parietal electrodes (PO3, PO4, PO7, PO8, P7, P8). N450 was computed using a time window of 150 ms (350–500 ms) at centro-parietal electrodes (CP3, CP4, P5, P6). SP was scored as the mean amplitude at centro-parietal electrodes (Cz, CPz, Pz) during a time window of 300 ms (650–950 ms).

### Multivariate pattern analysis

MVPA was based on support vector machine (SVM), a supervised machine learning method that operated on labeled samples. A binary label with 1 for the single task and –1 for the dual task was used in this study. The classification process at a given time point included a training phase and a testing phase. During the training step, the SVM would find a decision boundary that separated the samples in the input features using class labels. Next, the trained classifier is used to predict the class labels of new examples. For the spectral MVPA, we defined three frequency regions of interest (F-ROIs): theta (4–7 Hz), alpha (8–13 Hz), and beta (14–25 Hz). Then, three spatial regions of interest (S-ROIs) were chosen: fronto-central (Fz, FCz, Cz), occipital-parietal (POz, Oz, PO3, PO4, PO7, PO8, O1, O2), and central (C3, CP3, Cz, CP4, C4) to measure theta, alpha, and beta power, respectively (*Zhao et al.,*

*2015*; *Li et al., 2021*). For each frequency band, the selected features were the power difference between congruent and incongruent trials in the single task and dual task in every 20 ms of each ROI (see 'Time-frequency analysis' section for the power transformation method). All data were normalized and mapped to the range [–1, 1] within each class to mitigate scale effects.

A linear kernel SVM was chosen and trained through the LIBSVM toolbox (*Chang and Lin, 2011*). The linear SVM has only one parameter C that determines the trade-off between allowing misclassifications and training error minimization. The functions were as follows:

$$K(X_i, X_j) = X_i^T X_j$$

$$\min \frac{1}{2} \|\omega\|^2 + C \sum_{i=1}^{m} \xi(i)$$

A classifier was trained and tested at each time point by using a fivefold cross-validation procedure. Trials were randomly divided into five equal-sized folds. Then, a leave-one-out procedure was run in which the classifier was trained on four folds and tested on the remaining fold. In this case, different C values were tried within a certain range to find the best cross-validation accuracy (or the smallest value if there were more than one). We set the range of C values to base 2, and the exponent range to [–10, 10] with a step size of 0.2. Here, accuracy was used to quantify the performance of the classifier based on the results of cross-validation.

For statistical analyses, a nonparametric statistical test based on the cluster-level permutation was implemented to deal with multiple comparisons while maintaining strict control of the false alarm rate (*Maris and Oostenveld, 2007*). In detail, t-tests against chance level (acc = 0.5) were carried out at each data point to obtain the statistic values. Statistical values with p-values<0.01 were clustered together by summing their statistic values. The cluster alpha was set to 0.01 to reduce the likelihood of large clusters spanning the entire dataset (*Mensen and Khatami, 2013*). Additionally, to avoid using extremely transient intervals that may not capture meaningful psychological phenomena, we specified that at least two significant time points formed a cluster. Then, we permuted the data for 1000 times by randomly swapping condition labels to obtain a permutation distribution of the maximum cluster-level statistic under the null hypothesis of no condition difference. If there was any cluster with its real statistic larger than the threshold, we rejected the null hypothesis and concluded that there was a significant difference.

## Time-frequency analysis

To avoid edge artifacts, EEG data were re-epoched from –1000 to 2000 ms relative to stimulus onset. EEG data were converted into time-frequency domain data using continuous wavelet transform in Letswave software (https://www.letswave.org/; *Mouraux and Iannetti, 2008*). The cmor1-1.5 was chosen as the mother wavelet function for the time-frequency representations ranging from 1 to 30 Hz with a step size of 0.29 Hz. The single-trial time-frequency representation was obtained after wavelet transform, and they were then averaged throughout all trials per condition. The baseline normalizations were performed at each time point for each frequency using an event-related spectral perturbation transformation [$ER_{t,f}\% = [A_{t,f} – R_f]/R_f$] (*Pfurtscheller and Lopes da Silva, 1999*). Considering edge effects, duration requirements, and the need to exclude other influences (*Cohen, 2014*), a 400–200 ms pre-stimulus time window was selected as the baseline time window. For the statistical test, we only analyzed the frequency bands and corresponding time windows that could successfully classify different tasks in MVPA, namely theta (740–820 ms) and beta (920–1040 ms).

## Representational similarity analysis

Our RSA aimed to explore the correlation between neural interaction patterns and behavioral interaction patterns. First, we extracted data that could reflect interaction patterns, namely Stroop effect differences between single-task and dual-task conditions based on the subject level in RT, SP, theta, and beta. Prior to analysis, data were normalized by z-scoring the values across all subjects. We then constructed representational dissimilarity matrices (RDMs) measuring the geometrical distances between all subjects, yielding 32×32 symmetrical matrices. Here, Euclidean distance was employed to measure the distance between subjects (*Edelman, 1998*). Finally, we fitted a linear regression with

the neural RDMs as regressors and the behavioral RDM as the dependent variable to test which neural RDMs could predict behavioral RDMs. p-Values were corrected by the false discovery rate.

## Acknowledgements

This work was supported by grants from the National Natural Science Foundation of China (32371105).

## Additional information

### Funding

| Funder | Grant reference number | Author |
|---|---|---|
| National Natural Science Foundation of China | 32371105 | Antao Chen |

The funders had no role in study design, data collection and interpretation, or the decision to submit the work for publication.

### Author contributions

Yafen Li, Conceptualization, Data curation, Formal analysis, Visualization, Methodology, Writing – original draft, Writing – review and editing; Yixuan Lin, Conceptualization, Visualization, Methodology, Writing – review and editing; Qing Li, Conceptualization, Validation, Methodology; Yongqiang Chen, Zhifang Li, Validation, Methodology; Antao Chen, Conceptualization, Supervision, Writing – review and editing

### Author ORCIDs

Yafen Li ⓘ https://orcid.org/0009-0009-9853-7546
Yixuan Lin ⓘ https://orcid.org/0000-0001-7861-4035
Qing Li ⓘ https://orcid.org/0009-0004-1528-1826
Yongqiang Chen ⓘ https://orcid.org/0009-0005-1825-2622
Zhifang Li ⓘ https://orcid.org/0009-0008-9723-2655
Antao Chen ⓘ https://orcid.org/0000-0001-9321-681X

### Ethics

This study was conducted in accordance with the ethical standards outlined in the Declaration of Helsinki. Informed consent was obtained from all participants, and consent to publish was also obtained. The study was approved by the Human Research Human Ethics Committee of the Shanghai University of Sport, approval number 102772024RT005. The informed consent process and the ethical considerations of this study were outlined in the Methods section.

Reviewer #1 (Public review): https://doi.org/10.7554/eLife.100918.3.sa1
Reviewer #2 (Public review): https://doi.org/10.7554/eLife.100918.3.sa2
Author response https://doi.org/10.7554/eLife.100918.3.sa3

## Additional files

### Supplementary files

MDAR checklist

### Data availability

Data and codes presented in this work are available on OSF.

The following dataset was generated:

| Author(s) | Year | Dataset title | Dataset URL | Database and Identifier |
|---|---|---|---|---|
| Li Y | 2024 | Concurrent working memory load eliminates the Stroop effect | https://osf.io/89bpz/ | Open Science Framework, 10.17605/OSF.IO/89BPZ |

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
