## [Editor Report · eLife Assessment]

This **important** study investigates how working memory load influences the Stroop effect from a temporal dynamics perspective. **Convincing** evidence is provided that the working memory load influences the Stroop effect in the late-stage stimulus-response mapping instead of the early sensory stage. This study will be of interest to both neuroscientists and psychologists who work on cognitive control.

---

## [Referee Report · Reviewer #1 (Public review)]

Summary:

This study investigates an intriguing question in cognitive control from a temporal dynamics perspective: why does concurrent verbal working memory load eliminate the color-word Stroop effect? Through a series of thorough data analyses, the authors propose that verbal working memory load occupies the stimulus-response mapping resources represented by theta-band activity, thereby disrupting the mapping process for task-irrelevant distractors. This reduces the response tendency to the distractors, ultimately leading to the elimination of the Stroop effect.

Strengths:

The behavioral and neural evidence presented in the manuscript is solid, and the findings have valuable theoretical implications for research on Stroop conflict processing.

Comments on revisions:

The authors have addressed all concerns.

---

## [Referee Report · Reviewer #2 (Public review)]

Summary

Li et al. explored which stage of Stroop conflict processing was influenced by working memory loads. Participants completed a single task (Stroop task) and a dual task (the Sternberg working memory task combined with the Stroop task) while their EEG data was recorded. They adopted the event-related potential (ERP), and multivariate pattern analyses (MVPA) to investigate the interaction effect of task (single/dual) and congruency (congruent/incongruent). The results showed that the interaction effect was significant on the sustained potential (SP; 650-950 ms), the late theta (740-820 ms), and beta (920-1040 ms) power but not significant on the early P1 potential (110-150 ms). They used the representational similarity analyses (RSA) method to explore the correlation between behavioral and neural data, and the results revealed a significant contribution of late theta activity.

Strength

The experiment is well designed.

The data were analyzed in depth from both time and frequency domain perspectives by combining several methods.

Comments on revisions:

All my concerns have been properly addressed, no further comments.

---

## [Author Response]

The following is the authors’ response to the original reviews.

**Reviewer #1 (Public review):**
Comment 1: In the Results section, the rationale behind selecting the beta band for the central (C3, CP3, Cz, CP4, C4) regions and the theta band for the fronto-central (Fz, FCz, Cz) regions is not clearly explained in the main text. This information is only mentioned in the figure captions. Additionally, why was the beta band chosen for the S-ROI central region and the theta band for the S-ROI fronto-central region? Was this choice influenced by the MVPA results?

We thank the reviewer for the question regarding the rationale for the S-ROI selection in our study. The beta band was chosen for the central region due to its established relevance in motor control (Engel & Fries, 2010), movement planning (Little et al., 2019) and motor inhibition (Duque et al., 2017). The fronto-central theta band (or frontal midline theta) was a widely recognized indicator in cognitive control research (Cavanagh & Frank, 2014), associated with conflict detection and resolution processes. Moreover, recent empirical evidence suggested that the fronto-central theta reflected the coordination and integration between stimuli and responses (Senoussi et al., 2022). Although we have described the cognitive processes linked to these different frequencies in the introduction and discussion sections, along with the potential patterns of results observed in Stroop-related studies, we did not specify the involved cortical areas. Therefore, we have specified these areas in the introduction to enhance the clarity of the revised version (in the fourth paragraph of the Introduction section).

Regarding whether the selection of S-ROIs was influenced by the MVPA results, we would like to clarify here that we selected the S-ROIs based on prior research and then conducted the decoding analysis. Specifically, we first extracted the data representing different frequency indicators (three F-ROIs and three S-ROIs) as features, followed by decoding to obtain the MVPA results. Subsequently, the time-frequency analysis, combined with the specific time windows during which each frequency was decoded, provided detailed interaction patterns among the variables for each indicator. The specifics of feature selection are described in the revised version (in the first paragraph of the Multivariate Pattern Analysis section).

Comment 2: In the Data Analysis section, line 424 states: “Only trials that were correct in both the memory task and the Stroop task were included in all subsequent analyses. In addition, trials in which response times (RTs) deviated by more than three standard deviations from the condition mean were excluded from behavioral analyses.” The percentage of excluded trials should be reported. Also, for the EEG-related analyses, were the same trials excluded, or were different criteria applied?

We thank the reviewer for this suggestion. Beyond the behavioral exclusion criteria, trials with EEG artifacts were also excluded from the data for the EEG-related analyses. We have now reported the percentage of excluded trials for both behavioral and EEG data analyses in the revised version (in the second paragraph of the EEG Recording and Preprocessing section and the first paragraph of the Behavioral Analysis section).

Comment 3: In the Methods section, line 493 mentions: “A 400-200 ms pre-stimulus time window was selected as the baseline time window.” What is the justification in the literature for choosing the 400-200 ms pre-stimulus window as the baseline? Why was the 200-0 ms pre-stimulus period not considered?

We thank the reviewer for this question and would like to provide the following justification. First, although a baseline ending at 0 ms is common in ERP analyses, it may not be suitable for time-frequency analysis. Due to the inherent temporal smoothing characteristic of wavelet convolution in time-frequency decomposition, task-related early activities can leak into the pre-stimulus period (before 0 ms) (Cohen, 2014). This means that extending the baseline to 0 ms will include some post-stimulus activity in the baseline window, thereby increasing baseline power and compromising the accuracy of the results. Second, an ideal baseline duration is recommended to be around 10-20% of the entire trial of interest (Morales & Bowers, 2022). In our study, the epoch duration was 2000 ms, making 200-400 ms an appropriate baseline length. Third, given that the minimum duration of the fixation point before the stimulus in our experiment was 400 ms, we chose the 400 ms before the stimulus as the baseline point to ensure its purity. In summary, considering edge effects, duration requirements, and the need to exclude other influences, we selected a baseline correction window of -400 to -200 ms. To enhance the clarity of the revised version, we have provided the rationale for the selected time windows along with relevant references (in the first paragraph of the Time-frequency analysis section).

Comment 4: Is the primary innovation of this study limited to the methodology, such as employing MVPA and RSA to establish the relationship between late theta activity and behavior?

We thank the reviewer for this insightful question and would like to clarify that our research extends beyond mere methodological innovation; rather, it utilized new methods to explore novel theoretical perspectives. Specifically, our research presents three levels of innovation: methodological, empirical, and theoretical. First, methodologically, MVPA overcame the drawbacks of traditional EEG analyses based on specific averaged voltage intensities, providing new perspectives on how the brain dynamically encoded particular neural representations over time. Furthermore, RSA aimed to identify which indicators among the decoded were directly related to behavioral representation patterns. Second, in terms of empirical results, using these two methods, we have identified for the first time three EEG markers that modulate the Stroop effect under verbal working memory load: SP, late theta, and beta, with late theta being directly linked to the elimination of the behavioral Stroop effect. Lastly, from a theoretical perspective, we proposed the novel idea that working memory played a crucial role in the late stages of conflict processing, specifically in the stimulus-response mapping stage (the specific theoretical contributions are detailed in the second-to-last paragraph of the Discussion section).

Comment 5: On page 14, lines 280-287, the authors discuss a specific pattern observed in the alpha band. However, the manuscript does not provide the corresponding results to substantiate this discussion. It is recommended to include these results as supplementary material.

We thank the reviewer for this suggestion. We added a new figure along with the corresponding statistical results that displayed the specific result patterns for the alpha band (Supplementary Figure 1).

Comment 6: On page 16, lines 323-328, the authors provide a generalized explanation of the findings. According to load theory, stimuli compete for resources only when represented in the same form. Since the pre-memorized Chinese characters are represented semantically in working memory, this explanation lacks a critical premise: that semantic-response mapping is also represented semantically during processing.

We thank the reviewer for this insightful suggestion. We fully agree with the reviewer’s perspective. As stated in our revised version, load theory suggests that cognitive resources are limited and dependent on a specific type (in the second paragraph of the Discussion section). The previously memorized Chinese characters are stored in working memory in the form of semantic representations; meanwhile the stimulus-response mapping should also be represented semantically, leading to resource occupancy. We have included this logical premise in the revised version (in the third-to-last paragraph of the Discussion section).

Comment 7: The classic Stroop task includes both a manual and a vocal version. Since stimulus-response mapping in the vocal version is more automatic than in the manual version, it is unclear whether the findings of this study would generalize to the impact of working memory load on the Stroop effect in the vocal version.

We fully agree with the reviewer’s point that the verbal version of the Stroop task differs from the manual version in terms of the degree of automation in the stimulus-response mapping. Specifically, the verbal version relies on mappings that are established through daily language use, while the manual version involves arbitrary mappings created in the laboratory. Therefore, the stimulus-response mapping in the verbal response version is more automated and less likely to be suppressed. However, our previous research indicated that the degree of automation in the stimulus-response mapping was influenced by practice (Chen et al., 2013). After approximately 128 practice trials, semantic conflict almost disappears, suggesting that the level of automation in stimulus-response mapping for the verbal Stroop task is comparable to that of the manual version (Chen et al., 2010). Given that participants in our study completed 144 practice trials (in the Procedure section), we believe these findings can be generalized to the verbal version.

Comment 8: While the discussion section provides a comprehensive analysis of the study’s results, the authors could further elaborate on the theoretical and practical contributions of this work.

We thank the reviewer for the constructive suggestions. We recognize that the theoretical and practical contributions of the study were not thoroughly elaborated in the original manuscript. Therefore, we have now provided a more detailed discussion. Specifically, the theoretical contributions focus on advancing load theory and highlighting the critical role of working memory in conflict processing. The practical contributions emphasize the application of load theory and the development of intervention strategies for enhancing inhibitory control. A more detailed discussion can be found in the revised version (in the second-to-last paragraph of the Discussion section).

**Reviewer #2 (Public review):**
Comment 1: As the researchers mentioned, a previous study reported a diminished Stroop effect with concurrent working memory tasks to memorize meaningless visual shapes rather than memorize Chinese characters as in the study. My main concern is that lower-level graphic processing when memorizing visual shapes also influences the Stroop effect. The stage of Stroop conflict processing affected by the working memory load may depend on the specific content of the concurrent working memory task. If that’s the case, I sense that the generalization of this finding may be limited.

We thank the reviewer for this insightful concern. As mentioned in the manuscript, this may be attributed to the inherent characteristics of Chinese characters. In contrast to English words, the processing of Chinese characters relies more on graphemic encoding and memory (Chen, 1993). Therefore, the processing of line patterns essentially occupies some of the resources needed for character processing, which aligns with our study’s hypothesis based on dimensional overlap. Additionally, regarding the results, even though the previous study presents lower-level line patterns, the results still showed that the working memory load modulated the later theta band. We hypothesize that, regardless of the specific content of the pre-presented working memory load, once the stimulus disappears from view, these loads are maintained as representations in the working memory platform. Therefore, they do not influence early perceptual processing, and resource competition only occurs once the distractors reach the working memory platform. Lastly, previous study has shown that spatial loads, which do not overlap with either the target or distractor dimensions, do not influence conflict effect (Zhao et al., 2010). Taken together, we believe that regardless of the specific content of the concurrent working memory tasks, as long as they occupy resources related to irrelevant stimulus dimensions, they can influence the late-stage processing of conflict effect. Perhaps our original manuscript did not convey this clearly, so we have rephrased it in a more straightforward manner (in the second paragraph of the Discussion section).

Comment 2: The P1 and N450 components are sensitive to congruency in previous studies as mentioned by the researchers, but the results in the present study did not replicate them. This raised concerns about data quality and needs to be explained.

We thank the reviewer for this insightful concern. For P1, we aimed to convey that the early perceptual processing represented by P1 is part of the conflict processing process. Therefore, we included it in our analysis. Additionally, as mentioned in the discussion, most studies find P1 to be insensitive to congruency. However, we inappropriately cited a study in the introduction that suggested P1 shows differences in congruency, which is among the few studies that hold this perspective. To prevent confusion for readers, we have removed this citation from the introduction.

As for N450, most studies have indeed found it to be influenced by congruency. In our manuscript, we did not observe a congruency effect at our chosen electrodes and time window. However, significant congruency effects were detected at other central-parietal electrodes (CP3, CP4, P5, P6) during the 350-500 ms interval. The interaction between task type and consistency remained non-significant, consistent with previous results. Furthermore, with respect to the location of the electrodes chosen, existing studies on N450 vary widely, including central-parietal electrodes and frontal-central electrodes (for a review, see Heidlmayr et al., 2020). We speculate that this phenomenon may be related to the extent of practice. With fewer total trials, the task may involve more stimulus conflicts, engaging more frontal brain areas. On the other hand, with more total trials, the task may involve more response conflicts, engaging more central-parietal brain areas (Chen et al., 2013; van Veen & Carter, 2005). Due to the extensive practice required in our study, we identified a congruency N450 effect in the central-parietal region. We apologize for not thoroughly exploring other potential electrodes in the previous manuscript, and we have revised the results and interpretations regarding N450 accordingly in the revised version (in the N450 section of the ERP results and the third paragraph of the Discussion section).

**Recommendations for the authors:**

**Reviewer #1 (Recommendations for the authors):**
Comment 1: In the Introduction, line 108 states: “Second, alpha oscillations (8-13 Hz) can serve as a neural inverse index of mental activity or alertness, while a decrease in alpha power reflects increased alertness or enhanced attentional inhibition of distractors (Arakaki et al., 2022; Tafuro et al., 2019; Zhou et al., 2023; Zhu et al., 2023).” Please clarify which specific psychological process related to conflict processing is reflected by alpha oscillations.

We appreciate your suggestion and we have clearly highlighted the role of alpha oscillations in attentional engagement during conflict processing in the revised version (in the third-to-last paragraph of the introduction).

Comment 2: In Figures 3C and 3E, a space is needed between “amplitude” and the preceding parenthesis. Similar adjustments are required in Figures 4A, 4B, 4C, 5C, and 6C. Additionally, in Figures 3B and 3D, a space should be added between the numbers and “ms.” This issue also appears in Figure 8. Please review all figures for these formatting inconsistencies.

We apologize for the inconsistency in formatting and have corrected them throughout the revised version.

Comment 3: There are some clerical errors in the manuscript that need correction. For instance, on page 19, line 403: “Participants were asked to answer by pressing one of two response buttons (“S with the left ring finger and “L” with the left ring finger).” This should be corrected to: “L” with the right ring finger. I recommend that the authors carefully proofread the manuscript to identify and correct such errors.

We sincerely apologize for the errors present in the manuscript and have now carefully proofread it (in the Procedure section).

Comment 4: On page 13, line 254, the elimination of the Stroop effect should not be interpreted as an improvement in processing.

We greatly appreciate your suggestion. We agree that the elimination of the Stroop effect should not be confused with improvements in processing. We have corrected this in the revised version (the second paragraph of the Discussion section).

**Reviewer #3 (Recommendations for the authors):**
Comment 1: In the introduction section, the N450 was introduced as “a frontal-central negative deflection”, but in the methods part the N450 was computed using central-parietal electrodes. This inconsistency is confusing and needs to be clarified.

We apologize for this confusion. We have provided a detailed explanation regarding the differences in electrodes and the rationale behind choosing central-parietal electrodes in our response to Reviewer 2’s second comment. To clarify, we have updated the introduction to consistently label them as central-parietal deflections (in the third paragraph of the Introduction section).

Comment 2: I speculate the “beta” was mistakenly written as “theta” in line 212.

We sincerely apologize for this mistake. We have corrected this error (in the RSA results section).

Comment 3: The speculation that “changes in beta bands may be influenced by theta bands, thereby indirectly influencing the behavioral Stroop effect” needs to be rationalized.

We appreciate your suggestion. What we intended to convey is that we found an interaction effect in the beta bands; however, the RSA results did not show a correlation with the behavioral interaction effect. We speculate that beta activity might be influenced by the theta bands. On the one hand, we realize that the idea of beta bands indirectly influencing the behavioral Stroop effect was inappropriate, and we have removed this point in the revised version. On the other hand, we have provided rational evidence for the idea that beta bands may be influenced by theta bands. This is based on the biological properties of theta oscillations, which support communication between different cortical neural signals, and their functional role in integrating and transmitting task-relevant information to response execution (in the third-to-last paragraph of the Discussion section).

Comment 4: Typo in line 479: [10,10].

We sincerely apologize for this mistake. We have corrected this error: [-10,10] (in the Multivariate pattern analysis section).

Reference

Cavanagh, J. F., & Frank, M. J. (2014). Frontal theta as a mechanism for cognitive control. *Trends in Cognitive Sciences*, *18*(8), 414–421. https://doi.org/10.1016/j.tics.2014.04.012

Chen, M. J. (1993). A Comparison of Chinese and English Language Processing. In *Advances in Psychology* (Vol. 103, pp. 97–117). North-Holland. https://doi.org/10.1016/S0166-4115(08)61659-3

Chen, X. F., Jiang, J., Zhao, X., & Chen, A. (2010). Effects of practice on semantic conflict and response conflict in the Stroop task. *Journal of Psychological Science*, *33*, 869–871 (in Chinese).

Chen, Z., Lei, X., Ding, C., Li, H., & Chen, A. (2013). The neural mechanisms of semantic and response conflicts: An fMRI study of practice-related effects in the Stroop task. *NeuroImage*, *66*, 577–584. https://doi.org/10.1016/j.neuroimage.2012.10.028

Cohen, M. X. (2014). Analyzing Neural Time Series Data: Theory and Practice. The MIT Press. https://doi.org/10.7551/mitpress/9609.001.0001

Duprez, J., Gulbinaite, R., & Cohen, M. X. (2020). Midfrontal theta phase coordinates behaviorally relevant brain computations during cognitive control. *NeuroImage*, *207*, 116340. https://doi.org/10.1016/j.neuroimage.2019.116340

Duque, J., Greenhouse, I., Labruna, L., & Ivry, R. B. (2017). Physiological Markers of Motor Inhibition during Human Behavior. *Trends in Neurosciences*, *40*(4), 219–236. https://doi.org/10.1016/j.tins.2017.02.006

Engel, A. K., & Fries, P. (2010). Beta-band oscillations—Signalling the status quo? *Current Opinion in Neurobiology*, *20*(2), 156–165. https://doi.org/10.1016/j.conb.2010.02.015

Heidlmayr, K., Kihlstedt, M., & Isel, F. (2020). A review on the electroencephalography markers of Stroop executive control processes. *Brain and Cognition*, *146*, 105637. https://doi.org/10.1016/j.bandc.2020.105637

Little, S., Bonaiuto, J., Barnes, G., & Bestmann, S. (2019). Human motor cortical beta bursts relate to movement planning and response errors. *PLOS Biology*, *17*(10), e3000479. https://doi.org/10.1371/journal.pbio.3000479

Morales, S., & Bowers, M. E. (2022). Time-frequency analysis methods and their application in developmental EEG data. *Developmental Cognitive Neuroscience*, *54*, 101067. https://doi.org/10.1016/j.dcn.2022.101067

Senoussi, M., Verbeke, P., Desender, K., De Loof, E., Talsma, D., & Verguts, T. (2022). Theta oscillations shift towards optimal frequency for cognitive control. *Nature Human Behaviour*, *6*(7), Article 7. https://doi.org/10.1038/s41562-022-01335-5

van Veen, V., & Carter, C. S. (2005). Separating semantic conflict and response conflict in the Stroop task: A functional MRI study. *NeuroImage*, *27*(3), 497–504. https://doi.org/10.1016/j.neuroimage.2005.04.042

Zhao, X., Chen, A., & West, R. (2010). The influence of working memory load on the Simon effect. *Psychonomic Bulletin & Review*, *17*(5), 687–692. https://doi.org/10.3758/PBR.17.5.687